# A New Insight into the Composition and Physical Characteristics of Corncob—Substantiating Its Potential for Tailored Biorefinery Objectives

Pradeep Kumar Gandam [1], Madhavi Latha Chinta [2], A. Priyadarshini Gandham [3],
Ninian Prem Prashanth Pabbathi [1], Srilekha Konakanchi [4], Anjireddy Bhavanam [5], Srinivasa R. Atchuta [6], Rama Raju Baadhe [1,*] and Ravi Kant Bhatia [7,*]

1    Integrated Biorefinery Research Laboratory, Department of Biotechnology, National Institute of Technology Warangal, Warangal 506004, India
2    Stem Cell Research Laboratory, Department of Biotechnology, National Institute of Technology Warangal, Warangal 506004, India
3    Department of Pharmacology, CMR College of Pharmacy, Kandlakoya, Hyderabad 501401, India
4    Department of Biotechnology, Chaitanya (Deemed to be University), Hanamkonda 506001, India
5    Department of Chemical Engineering, NIT Jalandhar Campus, Jalandhar 144011, India
6    Center for Solar Energy Materials, International Advanced Research Center for Powder Metallurgy and New Materials, Balapur, Hyderabad 500005, India
7    Department of Biotechnology, Himachal Pradesh University, Summer Hill, Shimla 171005, India
*    Correspondence: rrb@nitw.ac.in (R.R.B.); ravibiotech07@hpuniv.ac.in (R.K.B.); Tel.: +91-83-3296-9462 (R.R.B.); +91-94-1827-2803 (R.K.B.)

**Abstract:** Corncobs of four different corn varieties were physically segregated into two different anatomical portions, namely the corncob outer (CO) and corncob pith (CP). The biomass composition analysis of both the CO and CP was performed by four different methods. The CP showed a higher carbohydrate and lower lignin content (83.32% and 13.58%, respectively) compared with the CO (79.93% and 17.12%, respectively) in all of the methods. The syringyl/guaiacyl (S/G) ratio was observed to be higher in the CP (1.34) than in the CO (1.28). The comprehensive physical characterization of both samples substantiated the lower crystallinity and lower thermal stability that was observed in the CP compared to the CO. These properties make the CP more susceptible to glycanases, as evident from the enzymatic saccharification of CP carried out with a commercial cellulase and xylanase in this work. The yields obtained were 70.57% and 88.70% of the respective theoretical yields and were found to be equal to that of pure cellulose and xylan substrates. These results support the feasibility of the tailored valorization of corncob anatomical portions, such as enzymatic production of xylooligosaccharides from CP without pretreatment combined with the bioethanol production from pretreated CO to achieve an economical biorefinery output from corncob feedstock.

**Keywords:** corncob anatomical portions; differential biomass composition analysis; crystallinity measurements; thermogravimetry; enzymatic saccharification without pretreatment; tailored biorefinery

## 1. Introduction

The increasing demand for biofuels has been the largest driving force for research and development in the field of biomass valorization [1]. The ubiquitous and continuous supply of agricultural waste has made it a promising biomass type for second-generation (2G) biofuels, or cellulosic biofuels, which are made from cellulose available from non-food crops and waste biomass such as corn stover, corncobs, straw, wood, and wood byproducts [2]. Maize (*Zea mays*) is the second most important cereal crop cultivated globally, with production reaching up to 1172.58 million metric tons (Mt) by the year 2022 [3]. This fact emphasizes the ready availability of corncobs, a unique xylan-rich agricultural waste generated during the processing of maize for its kernels. The anatomy of the corncob is constituted of diverse

physical components that can be broadly partitioned into an outer portion and an inner portion. The outer portion is a highly dense area comprising a woody ring, chaff, and beeswing. The inner portion that is soft and less dense is known as the pith (Figure 1). Corncob is one of the proven economical feedstocks for 2G biofuel production [1], with a comparatively high glucan and xylan content and a lower lignin percentage than the other agriculture-generated 2G feedstocks [4].

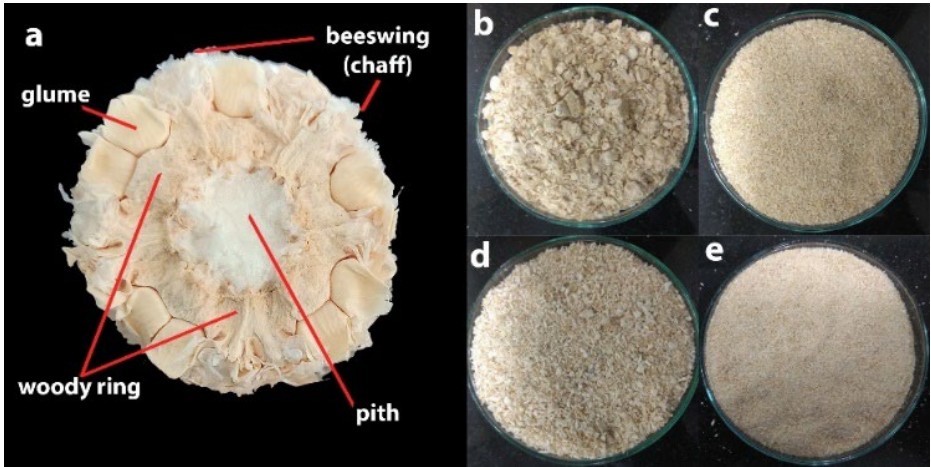

**Figure 1.** Corncob cross-sectional anatomy and the samples prepared. (**a**) corncob cross-section showing CO and CP regions; (**b**) CO comminuted to 2–10 mm; (**c**) CO comminuted to 0.85–0.18 mm (−20/+80 mesh); (**d**) CP comminuted to 2–5 mm; (**e**) CP comminuted to 0.85–0.18 mm (−20/+80 mesh).

The average lignocellulose composition of the whole corncob reported by several researchers is in the range of 33–43% cellulose, 26–36% hemicellulose, and 17–21% lignin [5,6]. These data invariably show the high hemicellulose composition of corncob compared with other biomass types. Unlike other biomass types where only 2G bioethanol is the main biorefinery product, the unique physicochemical construct of corncobs made it a feedstock of choice for many other value-added products. The high xylan content of corncobs has been used as a feedstock for the industrial-scale production of xylitol [7] and furfural [8]. A great deal of research has been reported for the production of xylooligosaccharides [9] and furan-derived biorefinery platforms such as furfurylamine [10], and furoic acid [11]. Corncob-derived sugars have been reported as the carbon source for the fermentative production of acids such as propionic acid [12], levulinic acid [13], lactic acid [14], acetic acid [14], butyric acid [15], malic acid [16], and alcohols such as ethanol [17], butanol [18], and 2,3-butanediol [19]. Further, the pretreated whole corncob meal was used as a carbon source for solid state and submerged fermentations [20], for the production of biogas [21], for the production of bio-hydrogen [22], and as a biosorbent to purify water by removing heavy metals [23,24], industrial dyes [25], and metal ions [26]. Other corncob-based products that have been reported, but are not limited to, are high-valued celluloses such as cellulose acetate [27], regenerated cellulose films [28], and the whole corncob pyrolysis-derived products [29]. In our recent review on corncob biorefineries, these products, their production routes, and the life cycle assessment studies of the corncob biorefinery were discussed in detail [30].

Biomass pretreatment has been the bottleneck in determining the overall productivity, economics, and life cycle energy consumption of any biorefinery [1,31]. A techno-economic analysis of corncob biorefineries inferred that the major stake in operating costs is on account of the biomass pretreatment process [1], which accounts for an average of 18% of the overall cost of biorefinery [32]. In addition to its direct cost, pretreatment shows a significant impact on both the upstream as well as downstream processes involved, such as the type of biomass used, sugar content in liquid fraction generated, choice of neutralization step, chosen organism to ferment the liquid fraction, ways to deal with

oligomers generated, quantities of ash, lignin, and extractives in liquid fraction, and their effect on enzymatic saccharification and fermentation, isolation of lignin and other inhibitors, methods to process the solid fraction, and the processes to deal with the waste and effluents. These manifestations are not only capital-intensive but also pose a significant impact on the environment. Several approaches have been proposed to minimize the overall operating cost of corncob biorefineries, such as further valorization of xylan or cellulose-extracted industrial corncob residues, co-utilization of corncob-derived glucan and xylan, and valorization of all three components of corncob-derived lignocellulose in a biorefinery fashion. These approaches explain the importance of co-product credit to make the overall process economically viable [1]. Majority of research on corncobs is focused on native farm-collected whole corncobs. Considerable research has been reported for the valorization of corncob waste residue (CCR) generated from corncob-derived industries as well [30]. Except for a few CCR valorization approaches that utilized CCR without a pretreatment [13,20], every other whole corncob biorefinery approach has been heavily invested in optimizing suitable pretreatment approaches [28]. Nevertheless, none of these approaches reported a scenario where tailored biochemical or thermochemical treatments were applied to individual anatomical portions of the corncob to achieve a better outcome.

The very idea of this current work is based on the belief that the corncob pith can be valorized with a mild pretreatment or without pretreatment, owing to its peculiar morphological features, to improve the overall economics of the biorefinery. To establish this, it is important to thoroughly understand the lignocellulosic construct and recalcitrance of these corncob anatomical portions.

Lignocellulose biomass recalcitrance is typically influenced by several chemical and physical factors. The chemical factors include composition (hemicellulose, cellulose, and lignin content), acetyl groups, hydroxyl groups, and syringyl/syringyl + guaiacyl (S/G) ratio. The physical parameters include crystallinity, degree of polymerization, accessible surface area, and accessible volume [33]. Lignin is known to cause unproductive binding with glycanases to prevent them from saccharifying the biomass [33]. This effect of lignin on glycanases is varied as per the innate abundance of its three monomers (syringyl (S), guaiacyl (G), and p-hydroxyl phenol (H)). Most research reports have claimed that a high S/G ratio of lignin favors enzymatic saccharification due to the relatively high affinity of G-subunits towards glycanases [34,35]. Enhanced syringyl content by genetically engineered plant cell walls showed lesser recalcitrance and higher susceptibility to enzymatic saccharification [36]. However, certain reports contradict the above assumption and state an opposite or no effect of the S/G ratio on enzymatic saccharification [37].

Crystallinity is the extensively studied supramolecular physical parameter of lignocellulose and pure cellulose materials, expressed as the ratio of the crystalline regions of the biomass to its amorphous regions. The close association of crystalline cellulose fibers with non-covalent interactions makes it around 3–30 times less susceptible to enzymatic hydrolysis than its amorphous regions [38]; most studies have reported the impeding effect of crystallinity on enzymatic saccharification [39]. However, again some reports have stated that crystallinity is comparatively less critical than other physical parameters, such as the degree of polymerization, particle size, pore volume, and accessible surface area, with respect to affecting the biomass recalcitrance [40]. Pretreatments have often been proven to achieve 10% more lignocellulose deconstruction with smaller biomass particles (<1 mm) than larger ones (1–4 mm) [41,42] and especially the highest lignin dissolution [43]. However, as the particle size decreases, sugar and solid recovery tend to decrease after pretreatment, showing a negative effect on downstream enzymatic saccharification and overall bioconversion. On the other hand, for processes such as biomass torrefaction and palletization, microparticle sizes are preferred over larger ones [44]. In addition, the process of biomass size reduction itself is an energy-intensive step; hence, a trade-off between the biomass particle size and the overall process economics must be empirically considered [41]. Accessible surface area or specific surface area (SA) is comparatively less studied but is a critical factor that determines enzymatic saccharification. SA is essentially related to the particle size and

pore volume of the biomass, where a reduction in the particle size or increase in the pore volume enhances the SA [39]. Some reports have stated that there is a threshold for particle size beyond which further comminution does not affect enzymatic saccharification, and these threshold particle sizes were observed to be different for each biomass type [45]. A typical cellulase molecular size is around 5.1 nm; hence, a lignocellulose pore volume large enough to fit a cellulase could theoretically enhance the scarification efficiency due to the percolation of the enzyme. A pore size range of 10–30 nm was reported to be effective for different biomass types to undergo enzymatic saccharification, and there are also studies that have reported a negative or no correlation at all for the SA of a biomass type to its enzymatic susceptibility [46]. Moreover, the proper empirical measurement of SA is always a difficult task [47].

All these findings invariably suggest that the recalcitrance of lignocellulosic biomass is a collective phenomenon that depends on all of the above-mentioned chemical and physical parameters put together rather than on any individual parameter. A great deal of the physical characterization of whole corncob through techniques such as scanning electron microscopy (SEM), X-ray diffraction (XRD), Fourier transform infrared spectroscopy (FTIR), Brunauer, Emmett, and Teller surface area analysis (BET) and thermogravimetric analysis (TGA) have been reported. Most studies have reported using at least more than one of these techniques to study the effect of corncob recalcitrance on its enzymatic saccharification, and they have showed that the decrease in biomass recalcitrance upon pretreatment promotes the enzymatic saccharification [48] in addition to all of the other types of biorefinery objectives of corncob discussed above [49].

To fill the aforementioned research gap, in this work we tried to establish the advantage and readiness of corncob anatomical portions for tailored biorefinery strategies by performing comprehensive compositional analysis, physical characterization, and enzymatic saccharification of the separated outer (CO) and pith (CP) portions of the corncob. We primarily focused on measuring crystalline and amorphous proportions and on establishing a detailed lignocellulosic composition of corncob anatomical portions to perceive their effect on enzymatic saccharification. More than one method was used for this study with the intent to make this work serve as a reference for future works, as the details had not been reported so far with respect to the individual anatomical portions of the corncob.

## 2. Materials and Methods

### 2.1. Sample Selection and Preparation

Four different *Zea mays* varieties (https://iimr.icar.gov.in/cultivars-2/, accessed on 30 November 2022), KMH-2589 (Kaveri seed company limited, Secunderabad, India, 500003), LTH 22 (Yaaganti Seeds Pvt. Ltd., Hyderabad, India, 500034), P3533 (Pioneer Hi-Bred Private Ltd., Hyderabad, India, 500081), and BL 900 (Bisco biosciences, Hyderabad, India, 500003), which were produced and cultivated around Telangana state, India (18.1124° N, 79.0193° E), were chosen for the study. These were termed CC1, CC2, CC3, and CC4, respectively. Five kilograms of shelled corncobs of each variety were directly collected from the fields, thoroughly washed, and air-dried for several months as per the National Renewable Energy Laboratory, USA-laboratory analytical procedure (NREL-LAP) [50]. The pith was separated from air-dried corncobs by drilling it out using a homogenizer motor attached with a high-speed steel (HSS) drill bit (twist bit) of a 6 mm size. The average weight ratio of the separated outer and inner anatomical portions of the corncob was 49, with densities of 403.6 kg/m³ and 128 kg/m³, respectively. These portions were separately milled to obtain a particle size in the range of 0.85–0.18 mm (−20/+80 sieve fraction) [51]. The woody ring of the corncob outer was more resilient to milling, and it required a heavy-duty knife mill to comminute it to the desired size. Two corncob-derived samples (−20/+80 fractions)—the corncob outer (CO), and corncob pith (CP) were considered for further biomass composition analysis (Figure 1). The CP is relatively homogenous, whereas the CO is a mix of chaff, glume, and woody ring. Hence, for biomass composition analysis by the NREL and near-infrared (NIR) spectroscopy-based rapid methods, sampling was performed by selecting 50 random 5 g selections from thoroughly

mixed individual CO and CP fractions of each corncob variety to achieve a uniform distribution of all anatomical variations among the samples. For physical characterization, single CO and CP samples that were an equal mix of all the corncob varieties used were selected.

Commercial microcrystalline cellulose (Avicel® PH-101, Sigma Aldrich, Burlington, MA, USA, 01805) and cellulose-cotton liters (Sigma Aldrich, Burlington, MA, USA, 01805) were taken as pure cellulose references. Lignin alkali (Sigma Aldrich, Burlington, MA, USA, 01805) and xylan from beech wood (Megazyme, Wicklow, Ireland, A98YV29) were used as pure lignin and xylan references. These were termed AC, CL, LG, and XY, respectively. Unless otherwise mentioned, all of the samples are processed in triplicates through all of the analytical procedures.

### 2.2. Scanning Electron Microscopy (SEM) Analysis

Morphological images of the samples were recorded with a scanning electron microscope (VEGA3 TESCAN LMU). Small amounts of dry individual samples (moisture < 1%) were fixed on to sample-holding stubs using carbon tape and were subjected to gold and palladium sputtering under vacuum (Gold Sputter Coater-SPI-MODULE). The SEM instrument was operated in secondary electrons detection mode with a 5–15 kV accelerating voltage and working distance of around 10 mm. Each sample was scanned at three different levels of magnification, ranging from 600× to 5000× [52].

### 2.3. NREL Method for Biomass Composition Analysis

The biomass composition analysis was carried out as per the NREL-LAPS (https://www.nrel.gov/bioenergy/biomass-compositional-analysis.html, accessed on 30 November 2022). The monosaccharides analysis was carried out using high-performance liquid chromatography (HPLC) (Prominence UFLC, Shimadzu, Kyoto, Japan, 604-8442) equipped with Rezex-RPM-monosaccharide-Lead (II) ion column (Phenomenex, Torrance, CA, USA, 90501-1430) and a suitable guard column. The HPLC analysis of acetate was performed using a Repromer-H (Dr. Maisch GmbH, Beim Brückle, Germany, 1472119) column along with an appropriate guard column. We ran 50 µL of the samples through the respective columns maintained at 80 °C in isocratic mode using HPLC-grade water as the mobile phase. The retention data were collected using a refractive index detector with flow cell temperature of 50 °C. Analysis of sucrose was carried out using a biochemistry analyzer (YSI-2950-D, Xylem, Washington, DC, USA, 20003) equipped with an immobilized enzyme membrane (YSI-2703). The standards used for all analytical procedures were HPLC-grade chemicals purchased from Sigma Aldrich, Burlington, MA, USA, 01805.

### 2.4. Van Soest Method for Fiber Analysis

Detergent partitioning of the fiber fraction of the lignocellulose materials followed by gravimetric analysis, which was proposed by Van Soest et al. [53], was used to determine the composition of the CO, CP, AC, and CL. Initially, neutral detergent fiber (NDF) (hemicellulose + cellulose + lignin + ash), acid detergent fiber (ADF) (cellulose + lignin + ash), and acid detergent lignin (ADL) (lignin) were determined among the samples. Further, the respective percentages of cellulose, hemicellulose, and lignin were gravimetrically calculated using Equations (1)–(3) [53]. The respective digestions were carried in 250 mL round bottom flasks in a heating mantle. And the filtration followed by drying and ashing was carried out in borosilicate filtration crucibles with grade-2 porosity.

$$Hemicellulose = NDF - ADF \qquad (1)$$

$$Cellulose = ADF - ADL \qquad (2)$$

$$Lignin = ADL \qquad (3)$$

### 2.5. NIR Spectroscopy Method for Rapid Biomass Composition Analysis

The NIR spectra of the CO and CP samples were collected in the diffuse reflection mode using a Cary Varian 5000-UV-Visible-NIR spectrophotometer, Agilent, Santa Clara, CA,

USA. The spectra were acquired by placing around 1 g of the sample in the powder cell at ambient temperature. Each sample was scanned in triplicates in the range of 1000 nm to 2500 nm, with 64 scans per spectrum. The average of the triplicate spectrum was considered for further analysis. Reflectance (R) data was converted to absorbance (A) using the equation $A = \log(1/R)$ [54]. An NIR calibration model with partial least squares regression (PLS) was built using the Unscrambler®-X software, version 10.4 (Aspen Technology, Inc., Bedford, MA, USA, 01730). Preprocessing of the spectral data was carried out using Savitzky-Golay smoothing and multiplicative scatter correction techniques. The PLS calibration models were built based on the full range of the spectrum, where two-thirds of the sample scans were taken as a reference set and the remaining scans were taken as the test set. Both sets were carefully selected to have equal representation from all four samples. The coefficient of multiple determination for calibration ($R^2(C)$), coefficient of multiple determination for validation ($R^2(V)$), coefficient of multiple determination for prediction ($R^2(P)$), standard error of calibration (SEC), standard error of prediction (SEP), and residual predictive deviation (RPD) are the important indicators used for the NIR-PLS model evaluation [54].

*2.6. Thermogravimetric Analysis (TGA)*

TGA (TGA 4000, Perkin Elmer, Waltham, MA, USA, 02451) of the samples was separately carried out in isothermal mode under an inert atmosphere (N2 flow around 19.8 mL/min), and oxidative atmosphere (air). The temperature range used was 30–800 °C at a constant heating rate of 200 °C/min. The TGA curve with mass percentage remaining against temperature was plotted using OriginPro2018 software, Ver.b9.5.1.195 (OriginLab Corporation, Northamton, MA, USA, 01060). The instrument-generated first derivative data was smoothened with the adjacent averaging method at 70-point smoothing, and the mass loss percentage per minute against temperature was plotted. This curve was used as an alternative to the derivative thermogram (DTG); hence, hereafter it is referred to as the DTG curve. The lignocellulosic composition of the samples was calculated using Equations (4)–(6). Their relative thermal degradation percentages were obtained from the respective TGA curves, where the inflection points were selected based on the corresponding superimposed DTG curve [55]. Additionally, the DTG curve is normalized and inverted by integrating the sample weight percentage at each time fraction of the derivative data ($m_i$) to the initial ($m_0$) and end ($m_\infty$) mass% of the sample using Equation (7) [56]. The peak deconvolution was separately performed on normalized DTG curves of both CO and CP by manually selecting the peaks at each devolatilization stage, and a multiple peak fit was performed using the Gaussian function. Peaks were manually marked and iterations were performed until the fit converged and a chi-square tolerance value of $1 \times 10^{-9}$ was reached. All the converged peaks have shown $R^2$ and adjusted $R^2$ values above 0.99. Moisture, hemicellulose, cellulose, and lignin peaks were assumed as pseudo-components [57], and their compositions were calculated based on the respective areas of the peaks using Equation (8).

$$\% \text{ Hemicellulose } = (W - H) \tag{4}$$

$$\% \text{ Celulose } = (A - C) \tag{5}$$

$$\% \text{ Lignin } = (C - L) \tag{6}$$

$$X_i = \frac{m_i - m_\infty}{m_0 - m_\infty} \tag{7}$$

$$\% \text{ PC} = (a / A) \times 100 \tag{8}$$

where: $W$ = % mass after dehydration; $H$ = % mass measured after hemicellulose removal; $C$ = % mass measured after cellulose removed; $L$ = % mass measured after lignin removed (% Ash content); $PC$ = pseudo-component; $a$ = area of a peak; $A$ = total area under the curve.

*2.7. Fourier Transform Infrared Spectroscopy (FTIR) Analysis*

FTIR spectra were measured using a BRUKER Alpha II compact FTIR spectrometer. Both the CO and CP samples were milled to pass through an 80-mesh sieve, and the commercial

control samples AC, CL, and LG were used in their manufactured form without any additional milling. The samples were prepared as per the standard KBr pelleting method [58]. Spectra were collected in the absorbance mode with 32 scans per spectrum at a resolution of 4 cm$^{-1}$, within a wavenumber range of 4000–400 cm$^{-1}$ [59]. Each sample was pelleted in triplicates and an average spectrum was considered. Processing, mathematical analysis, and deconvolution of the obtained spectra were performed using OriginPro2018 software. The total crystallinity index (TCI) was calculated as the height ratio of the absorption peaks at 1372 cm$^{-1}$ and 2900 cm$^{-1}$ [60]. The lateral order index (LOI) or empirical crystallinity index was calculated as the area ratio of the peaks at 1430 cm$^{-1}$ and 893 cm$^{-1}$ [61]. Hydrogen bond intensity (HBI) was calculated as the area ratio of the peaks around 3340–3330 cm$^{-1}$ and 1320 cm$^{-1}$ [62]. Additionally, two different S/G ratios 1462 cm$^{-1}$/1510 cm$^{-1}$ [63] and 1595 cm$^{-1}$/1509 cm$^{-1}$ [64], lignin to total carbohydrate ratios 1515 cm$^{-1}$/1374 cm$^{-1}$, 1515 cm$^{-1}$/1162 cm$^{-1}$, and 1515 cm$^{-1}$/898 cm$^{-1}$, and hemicellulose to total carbohydrate ratio 1734 cm$^{-1}$/1374 cm$^{-1}$ [65] were calculated. Unless otherwise mentioned, the areas of the respective peaks were used to calculate all of the above-mentioned ratios.

*2.8. X-ray Diffraction (XRD) Analysis*

XRD data of the samples were recorded with X'Pert Powder XRD (Malvern Panalytical Ltd., Malvern, U.K, WR141XZ). The scans were performed at a step size of 0.0167113 in the 2θ angle range of 6–80° with 5 s of exposure at each step using Ni-filtered Cu Kα radiation at wavelengths of 1.540598 (Kα1) and 1.544426 (Kα2). The operating generator voltage and tube currents were 45 kV and 30 mA, respectively. Smoothing, baseline subtraction, peak integration, and peak deconvolution of the digitally obtained diffraction data between the 2θ angles from 10° to 40° were performed using OriginPro2018 software. The crystallinity of the samples was calculated by four different methods. The percent crystallinity index (CrI%) was calculated by the peak height method using Equation (9) [66]. Percent crystallinity (Crd) was calculated by the peak deconvolution method using Equation (10). This method assumes that the peak broadening is contributed by the amorphous content [67]. The percent crystallinity of the sample (Cra$_1$) was calculated by the amorphous contribution subtraction method using the ball-milled AC as the amorphous standard for all of the samples using Equation (11) [68]. This method needs an additional normalization step to bring the diffractogram of the amorphous standard below the sample diffractogram to avoid negative values making the process prone to errors or bias [68]. To overcome this problem, we reported a modified version of the amorphous contribution subtraction method where the percent crystallinity (Cra$_2$%) was measured using the ball-milled form of the sample itself as an amorphous standard instead of a common standard. The crystallite sizes of the (002) lattice of each sample were calculated using the Scherr equation (Equation (12)) [69], and the interplanar distances between the crystal lattices, known as d-spacing, were calculated using Bragg's law (Equation (13)) [70].

$$CrI\% = \left( \frac{I_{002} - I_{am}}{I_{002}} \right) \times 100 \qquad (9)$$

$$Cr_d\% = \left( \frac{A_{cr}}{A_t} \right) \times 10 \qquad (10)$$

$$Cr_{a1}\% = \left( \frac{A_{Cra1}}{A_s} \right) \times 100 \qquad (11)$$

$$L = k\lambda / \beta \cos \theta \qquad (12)$$

$$d = n \lambda / (2\sin \theta) \qquad (13)$$

where $I_{002}$ = Intensity at about 2θ = 22.6° (represents the diffraction from both crystalline and amorphous materials) $I_{am}$ = Intensity at the "valley" between the two peaks at about 2θ = 18° (represents the diffraction contributed by amorphous material), $A_{cr}$ is the area of

all the crystalline peaks (($(101)$, $(10\bar{1})$, $(021)$, $(002)$, $(040)$)) together, and $A_t$ is the total area of the diffractogram. $A_{Cra1}$ is the area of all the crystalline peaks of the sample obtained by peak integration after subtracting the diffraction intensity of the ball-milled AC and $A_s$ is the total area of the sample before amorphous subtraction. L is the crystallite size in nm, k is the dimensionless shape factor (0.89), λ is the wavelength of the incident X-ray (0.1540 nm), β is the full width at the half maximum (FWHM) of the (002) lattice expressed in radians, θ is the peak position in radians (Bragg angle), and n is a positive integer.

### 2.9. Enzymatic Saccharification of Untreated Corncob Samples

Both the CO and CP were separately saccharified with cellulase (*Trichoderma reesei* ATCC 26921, Sigma-C2730, initial activity around 650 filter paper units (FPU)/g), and xylanase (endo-1,4-β-Xylanase M1 from Trichoderma viride, Megazyme, E-XYTR1, initial activity around 1650 units (U)/mL), without any pretreatment. The CL and XY were also saccharified as the substrate controls with the respective enzymes. A typical enzymatic reaction process involved a 5 g dry weight of the substrate, taken in 250 mL Erlenmeyer flasks along with 50 mM of sodium citrate buffer, pH 4.8 (cellulase reaction), and pH 4.5 (xylanase reaction). Each enzyme was appropriately diluted in their respective buffers to achieve 20 FPU of cellulase and 30 U of xylanase per 1 g of dry mass of the substrate, achieving a liquid-to-solid ratio of 20 at a total reaction volume of 100 mL. A set of substrate blanks were incubated along with the test flasks by including all the ingredients mentioned above except the respective enzymes. The reactions were carried at 50 °C with shaking at 130 RPM for 50 h. Sample aliquots of 0.05 mL were collected at every 5 h interval. All the aliquots were appropriately diluted with respective buffer solutions to measure the total reducing sugars released using a micro-DNS assay, where the total reaction volume was minimized to 1.5 mL while maintaining the sample-to-reagent ratio mentioned in the original macro-DNS assay, as proposed by T.K. Ghose [71]. The absorbance of substrate blanks was subtracted from that of the corresponding test sample of the same time interval, and the resulting spectral data were plotted against time to visualize the enzymatic saccharification effect on each substrate. Enzyme activity (saccharification) was measured as per the procedure reported by Asmarani et al. [72]. The obtained saccharification yield was expressed as the percent of the total theoretical yield (TY), calculated using the equation of Mandels and Sternberg [73]. Anhydro correction factors of 0.9 and 0.88 were used for the cellulase and xylanase activities, respectively [73], and the total glucan and xylan concentrations obtained from the NREL analysis were taken as the respective initial substrate concentrations [74].

## 3. Results and Discussion

### 3.1. SEM Analysis

The SEM images revealed the varied morphological features of the samples (Figure 2). The CO is compact and tightly packed in contrast to the loosely packed foam-like CP. The pores observed in the CP explain its soft airy features. A huge contrast in physical recalcitrance can be observed between the CO and CP at every magnification (50 μm, 20 μm, and 5 μm). Several previously reported studies described the morphology of whole corncob particles as a sheet-like bulky structure [75], solid-tight structure [48], highly ordered rigid structure [76], and agglomerated unbroken surface [77], and those findings exactly coincide with the morphology of the CO of this study. In addition, these reports also presented an increase in corncob porosity upon pretreatment.

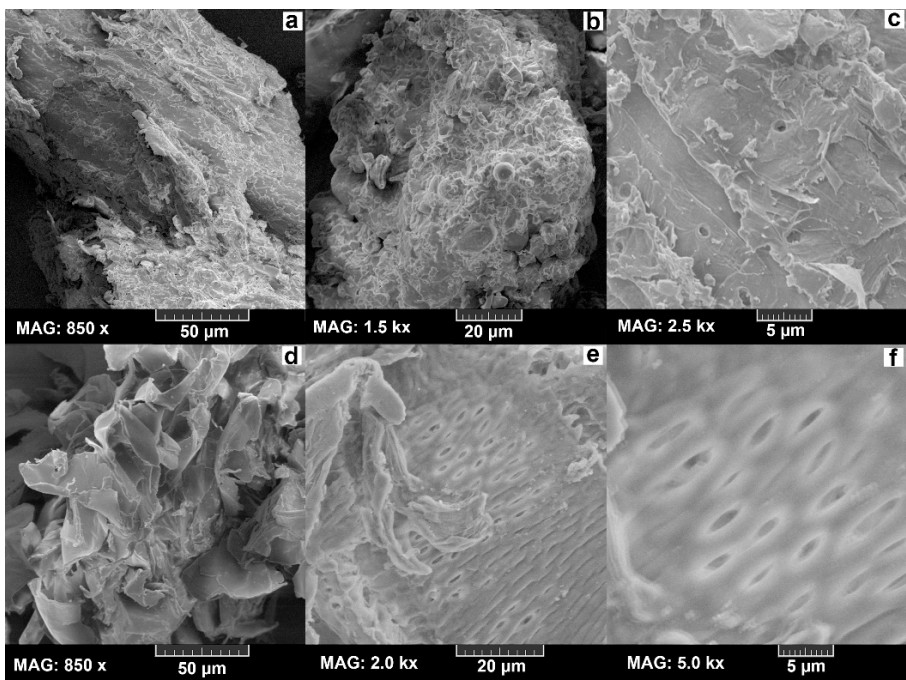

**Figure 2.** SEM images. Note: (**a**–**c**) are the CO and (**d**–**f**) are the CP. All the images were scanned at a constant accelerated voltage (H.V) of 5.0 kV by maintaining a working distance (W.D) ranging between 10.04 and 10.34 mm.

### 3.2. NREL Method for Biomass Composition Analysis

The compositional differences among all four different corncob varieties of the study were tabulated (Table 1). None of the CO and CP samples showed mannose, while a small percentage of mannose was found in both the CL and AC references. Both cellulose and hemicellulose percentages of all the CP samples were slightly greater than that of CO samples due to the comparatively lower total lignin percentage in the CP. Overall hemicellulose percentage among both the CO and CP samples was greater than the cellulose percentage (Table 1). The total water and ethanol extractives and the sucrose concentration in all CP samples were greater than that of the CO samples. The total protein was less in the CP than that of CO (Table 1). Many works reported biomass composition analysis of the whole corncob by the NREL method. However, most of these works reported just the cellulose, hemicellulose, and total lignin concentrations rather than the particulars of individual monosaccharide concentrations, the information about extractives, and the protein content. The lignocellulose composition of CO reported in this work is closer to that of the whole corncob composition reported in the literature [78], which could be due to the higher percentage of CO in the whole corncob.

The chromatograms related to calibration standards and the sample analysis are provided in the Supplementary Materials (Figures S1–S12).

### 3.3. Van Soest Method for Fiber Analysis

The NDF value of all CP samples was higher than that of CO and was similar to that of the pure cellulose references CL and AC. Although ADF values of CP were slightly higher than CO, they were almost half that of CL and AC. The composition analysis shows that the hemicellulose percentages of both the CO and CP samples were higher than their respective cellulose percentages. In addition, the CP samples showed comparatively higher cellulose and hemicellulose as well as lower lignin percentages compared with CO samples (Table 2). These results are consistent with the NREL method results reported in this work. Whole corncob fiber analysis results reported by many previous works [79] were closer to that of the CO in this work.

**Table 1.** Biomass composition of samples by the NREL method.

| Corn Variety/ Reference | Sample | %AIL | %ASL | %Glucan | %Xylan | %Galactan | %Arabinan | %Mannan | %Protein (Structural) | %Water Extractives | %Ethanol Extractives | %Sucrose | %Acetate |
|---|---|---|---|---|---|---|---|---|---|---|---|---|---|
| CC1 | CO | 14.52 ± 0.23 | 1.85 ± 0.13 | 36.68 ± 0.13 | 25.42 ± 0.26 | 10.1 ± 0.04 | 5.29 ± 0.26 | 0 ± 0.38 | 0.62 ± 0.1 | 2.26 ± 0.15 | 1.17 ± 0.22 | 2.58 ± 0.2 | 5.24 ± 0.38 |
| | CP | 11.11 ± 0.16 | 1.72 ± 0.12 | 39.13 ± 0.37 | 24.39 ± 0.34 | 11.14 ± 0.05 | 6.28 ± 0.28 | 0 ± 0.29 | 0.39 ± 0.13 | 3.49 ± 0.05 | 1.58 ± 0.04 | 3.84 ± 0.31 | 5.21 ± 0.07 |
| CC2 | CO | 15.44 ± 0.33 | 2.04 ± 0.31 | 37.04 ± 0.36 | 25.77 ± 0.19 | 11.45 ± 0.24 | 5.77 ± 0.06 | 0 ± 0.31 | 0.79 ± 0.05 | 2.46 ± 0.37 | 1.55 ± 0.15 | 2.89 ± 0.27 | 5.84 ± 0.2 |
| | CP | 11.18 ± 0.24 | 2.11 ± 0.35 | 39.66 ± 0.35 | 25.39 ± 0.1 | 11.52 ± 0.39 | 7.39 ± 0.12 | 0 ± 0.25 | 0.48 ± 0.13 | 3.59 ± 0.07 | 1.96 ± 0.25 | 4 ± 0.29 | 5.73 ± 0.19 |
| CC3 | CO | 14.52 ± 0.15 | 2.51 ± 0.12 | 37.22 ± 0.26 | 25.86 ± 0.1 | 10.63 ± 0.16 | 6.55 ± 0.12 | 0 ± 0.34 | 0.69 ± 0.37 | 2.28 ± 0.36 | 1.77 ± 0.39 | 2.87 ± 0.08 | 5.57 ± 0.2 |
| | CP | 11.42 ± 0.14 | 2.49 ± 0.37 | 40.44 ± 0.06 | 24.89 ± 0.17 | 11.26 ± 0.16 | 7.16 ± 0.05 | 0 ± 0.2 | 0.49 ± 0.32 | 3.35 ± 0.36 | 1.68 ± 0.28 | 4.19 ± 0.1 | 5.56 ± 0.13 |
| CC4 | CO | 15.52 ± 0.14 | 2.1 ± 0.26 | 37.71 ± 0.21 | 26.66 ± 0.09 | 11.65 ± 0.17 | 5.93 ± 0.03 | 0 ± 0.04 | 0.7 ± 0.14 | 2.85 ± 0.19 | 1.64 ± 0.29 | 2.76 ± 0.16 | 5.87 ± 0.33 |
| | CP | 12.04 ± 0.17 | 2.25 ± 0.11 | 39.64 ± 0.18 | 25.14 ± 0.34 | 12.15 ± 0.1 | 7.72 ± 0.33 | 0 ± 0.23 | 0.52 ± 0.07 | 3.37 ± 0.07 | 1.9 ± 0.27 | 4.21 ± 0.25 | 5.25 ± 0.3 |
| Reference | CL | 0.33 ± 0.27 | 0.35 ± 0.07 | 66.66 ± 0.24 | 15.47 ± 0.26 | N.D | N.D | 10.8 ± 0.2 | 0 ± 0.26 | 0.34 ± 0.07 | 0.25 ± 0.14 | 0 ± 0.15 | 0 ± 0.3 |
| | AC | 0 ± 0.04 | 0.32 ± 0.1 | 71.88 ± 0.11 | 15.83 ± 0.13 | N.D | N.D | 9.77 ± 0.36 | 0 ± 0.16 | 0.09 ± 0.34 | 0.07 ± 0.3 | 0 ± 0.25 | 0 ± 0.36 |

AIL: acid-insoluble lignin; ASL: acid-soluble lignin; N.D: not detected.

**Table 2.** Fiber analysis and lignocellulose composition analysis by the Van Soest method.

| Corn Variety/ Reference | Sample | % NDF | % ADF | % ADL | % Hemicellulose | % Cellulose | % Lignin |
|---|---|---|---|---|---|---|---|
| CC1 | CO | 87.17 ± 0.3 | 45.25 ± 0.14 | 6.75 ± 0.07 | 41.92 ± 0.07 | 38.5 ± 0.15 | 6.75 ± 0.1 |
| | CP | 92.76 ± 0.1 | 49.35 ± 0.16 | 1.7 ± 0.32 | 43.41 ± 0.16 | 47.65 ± 0.32 | 1.7 ± 0.12 |
| CC2 | CO | 85.56 ± 0.08 | 47.88 ± 0.1 | 9.47 ± 0.31 | 37.68 ± 0.3 | 38.41 ± 0.17 | 9.47 ± 0.15 |
| | CP | 95.62 ± 0.25 | 51.77 ± 0.22 | 4.12 ± 0.31 | 43.85 ± 0.13 | 47.65 ± 0.18 | 4.12 ± 0.11 |
| CC3 | CO | 88.02 ± 0.28 | 46.91 ± 0.3 | 9.34 ± 0.13 | 41.11 ± 0.24 | 37.57 ± 0.28 | 9.34 ± 0.09 |
| | CP | 94.43 ± 0.15 | 49.64 ± 0.24 | 2.36 ± 0.32 | 44.79 ± 0.24 | 47.28 ± 0.1 | 2.36 ± 0.11 |
| CC4 | CO | 86.21 ± 0.09 | 46.31 ± 0.19 | 8.3 ± 0.25 | 39.9 ± 0.27 | 38.01 ± 0.3 | 8.3 ± 0.1 |
| | CP | 95.1 ± 0.24 | 50.62 ± 0.21 | 1.8 ± 0.16 | 44.48 ± 0.12 | 48.82 ± 0.17 | 1.8 ± 0.19 |
| Reference | CL | 98.1 ± 0.31 | 95.51 ± 0.13 | 0 | 2.59 ± 0.28 | 95.51 ± 0.21 | 0 |
| | AC | 98.62 ± 0.17 | 97.31 ± 0.22 | 0 | 1.31 ± 0.11 | 97.31 ± 0.11 | 0 |

### 3.4. NIR Method for Rapid Biomass Composition Analysis

The NIR spectra of both the CO and CP were analogous to that of other biomass types reported [54], with all the characteristic peaks of lignocellulose. The results of PLS calibration, validation, and prediction performances of the individual models as per their full spectral pretreatment are presented in Figure 3. All the statistical parameters of both calibration and validation sets were similar. Among the models generated with the unprocessed spectra of CO, the glucan model achieved the highest prediction, followed by the models of sucrose and protein. Meanwhile, the highest predictive models of CP were obtained for xylan and protein, followed by sucrose, glucan, and lignin.

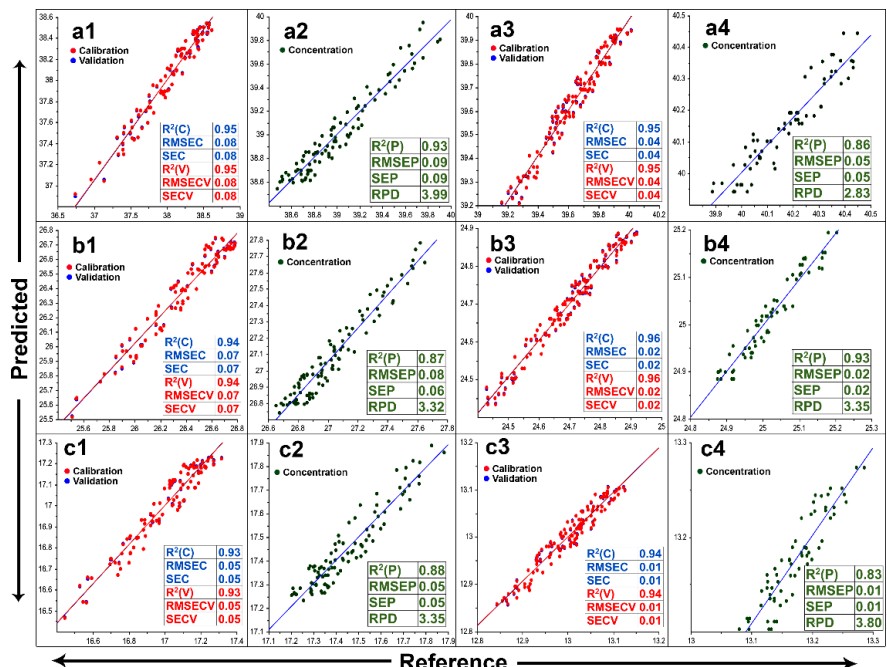

**Figure 3.** NIR-PLS calibration models. Note: (**a1**–**c1**) are calibration and validation models of the glucose, xylose, and lignin of CO, respectively; (**a2**–**c2**) are prediction performances of the models (**a1**–**c1**), respectively; (**a3**–**c3**) are calibration and validation models of the glucose, xylose, and lignin of CP, respectively; (**a4**–**c4**) are prediction performances of the models (**a3**–**c3**), respectively. Savitzky-Golay smoothing was used for the respective NIR spectra of all above models; $R^2$(C): coefficient of multiple determination for the calibration; $R^2$(V): coefficient of multiple determination for the validation; $R^2$(P): coefficient of multiple determination for the prediction; SEC: standard error of the calibration; SEP: standard error of the prediction; RPD: residual predictive deviation.

$R^2$(C)/$R^2$(P) ratios close to one, lower SEC and SEP values, and higher RPD values (>2) indicate a better fit of the models. The performances of all the models were significantly improved by the spectral pretreatments, decreasing the differences among calibration and

validation sets. Savitzky-Golay smoothing of both the CO and CP spectra achieved models with the highest predictive performance.

### 3.5. TGA Analysis

Under an inert environment, devolatilization started at 30 °C and maximum dehydration occurred between 50.5 and 67 °C. The end of the dehydration stage, denoted by the start of the first mass loss plateau, was observed in the range of 90.8–240 °C. An abrupt weight loss due to hemicellulose decomposition was observed at 298 °C for both the CO and CP [57], while the cellulose degradation peaks of the CO, CP, AC, and CL were in the range of 340–352 °C; the complete degradation of the same samples was in the range of 381–400 °C. No additional peaks were observed after 400 °C for all samples except for LG. In contrast, the thermal decomposition curve of all samples under the oxidative environment was comparatively complex, with additional devolatilization peaks observed at 423–472 °C for CO and CP, and around 591–598 °C for AC and CL. Maximum decomposition under the oxidative environment for CO and CP was achieved at 539 °C and 494 °C, respectively. The absence of a hemicellulose degradation peak in both AC and CL indicates their purity. The pyrolytic profile of LG under both inert and oxidative environments was quite complex with multiple decomposition steps, spanning a wide range of temperatures. Evidently, LG needs a temperature beyond 800 °C for complete decomposition. Both CO and CP achieved a higher mass loss under the oxidative environment. On the contrary AC, CL, and LG attained maximum weight loss under the inert environment (Figure 4). Despite showing similar degradation temperatures, the extent of pyrolysis among CO and CP is different, with CP showing a higher mass loss percentage at each inflection point. The three-stage thermal degradation profile of whole-native corncob reported by Yao et al. [80] is quite similar to that of the CO in this study, the starting, peak, and final temperatures of the TGA profile, including the maximum weight loss reported, were similar. The same is the case with the TGA of the whole corncob reported by Zheng et al. [81]. The alteration of the TGA profile reported for dilute sulfuric acid-pretreated corncob with that of native corncob showed the exact thermal decomposition temperature range of hemicellulose [81]. The lignocellulose composition of CO and CP calculated by the TGA analysis under both inert and oxidative environments clearly showed lower lignin and residue content along with a higher hemicellulose percentage in CP. The lignocellulose composition calculated as pseudo-components by the peak deconvolution method revealed a similar difference between CO and CP (Table 3, Figure 5). AC and CL have shown a pure cellulose devolatilization peak without traces of hemicellulose or lignin. These results are consistent with the compositions determined by the other methods reported in this work.

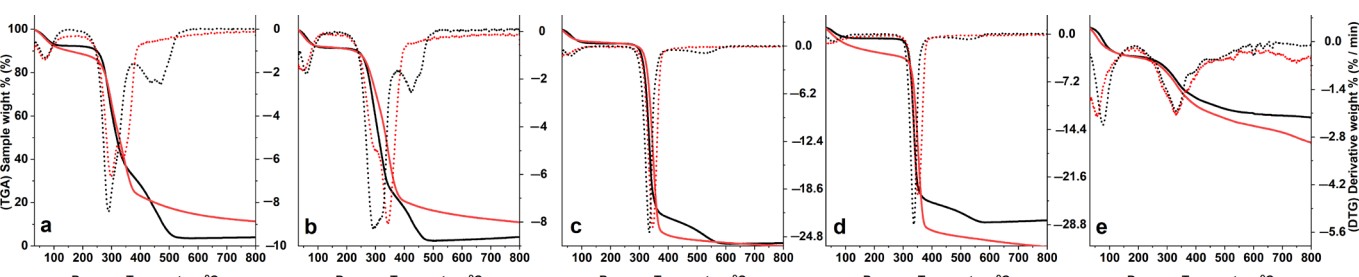

**Figure 4.** TGA profiles of the samples along with their first derivatives. Note: black solid and dotted lines: the thermogram and its derivative under the oxidative environment, respectively; red solid and dotted lines: the thermogram and its derivative under the inert environment, result; (**a**–**e**): CO, CP, AC, CL, and LG, respectively. The left-Y axis is common for all of the graphs.

**Table 3.** Mass (%) of the lignocellulose components in thermally degraded samples.

|  | CO-i | CO-o | CO-dc | CP-i | CP-o | CP-dc | AC-i | AC-o | CL-i | CL-o |
|---|---|---|---|---|---|---|---|---|---|---|
| HC | 24.23 | 24.97 | 25.31 | 29.93 | 32.83 | 45.09 | 0 | 0 | 0 | 0 |
| CE | 51.85 | 45.88 | 18.03 | 48.64 | 49.1 | 31.20 | 94.76 | 86.58 | 100 | 87.95 |
| LG | 12.15 | 24.99 | 16.58 | 10.09 | 13.91 | 13.16 | 5.24 | 12.01 | 0 | 9.09 |
| A and C | 11.35 | 4 | N.A | 10.9 | 4 |  | 0 | 1.37 | 0 | 2.9 |
| TC | 76.09 | 70.86 | 43.34 | 78.57 | 81.93 | 76.29 | 94.76 | 86.58 | 100 | 87.95 |
| HC/TC | 0.32 | 0.35 | 0.58 | 0.38 | 0.40 | 0.59 | 0 | 0 | 0 | 0 |
| LG/TC | 0.16 | 0.35 | 0.38 | 0.13 | 0.17 | 0.17 | 0.06 | 0.14 | 0 | 0.1 |

i: inert environment; o: oxidative environment; dc: peak deconvolution; HC: Hemicellulose; CE: cellulose; LG: lignin; TC: total carbohydrate; A and C: ash and residual carbon at 800 °C.

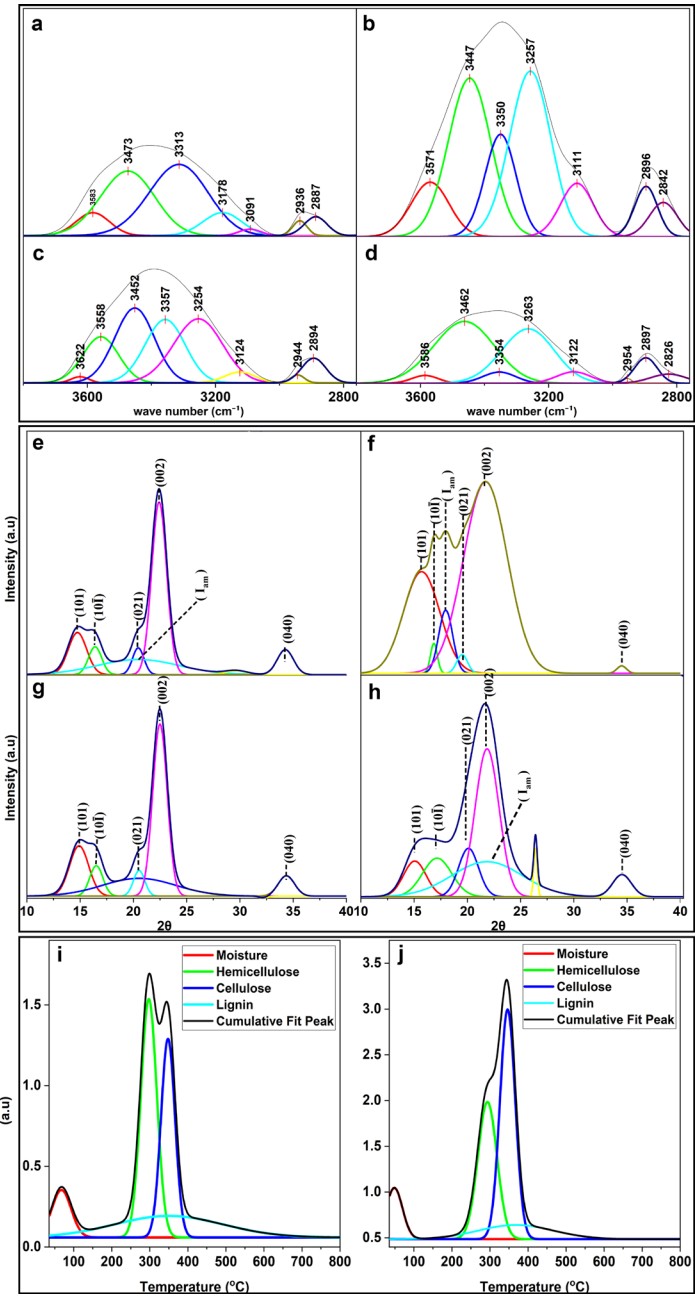

**Figure 5.** Peak deconvolutions of the FTIR, XRD, and DTG curves. Note: FTIR peak deconvolutions of (**a**) CO, (**b**) AC, (**c**) CP, and (**d**) CL; XRD peak deconvolutions of (**e**) CO, (**f**) AC, (**g**) CP, and (**h**) CL; DTG peak deconvolutions of (**i**) CO and (**j**) CP.

### 3.6. FTIR Analysis

The characteristic FTIR peaks of lignocellulose observed among all of the samples were tabulated (Table 4). The unprocessed spectra of all samples showed the characteristic -OH stretch in the range of 3700–3000 cm$^{-1}$, specifically at 3350 cm$^{-1}$ for both AC and CL and in the higher wavenumber region in the case of CO, CP, and LG. The -OH stretching peak of CP was much sharper and showed higher absorption than that of CO (Figure 5). Deconvolution of the broad stretching region between 3800 and 2800 cm$^{-1}$ showed around five different peaks for each sample (Figure 5). The relative peak intensities of the characteristic intramolecular hydrogen bonds (3586–3559 cm$^{-1}$, 3475–3448 cm$^{-1}$, and 3358–3351 cm$^{-1}$) were in the order of AC > CP > CO > CL, AC > CL > CO > CP, and AC > CP > CL, respectively. Furthermore, the intensities of intermolecular hydrogen bond peaks (3179–3112 cm$^{-1}$) were in the order of AC > CO > CL > CP. CP clearly showed an increased carbohydrate percentage compared with CO in both crystalline (1428 cm$^{-1}$, 1162 cm$^{-1}$) and amorphous regions (1335 cm$^{-1}$, 897 cm$^{-1}$, 668 cm$^{-1}$, 527 cm$^{-1}$, 993 cm$^{-1}$). In addition, CP showed an increased hemicellulose percentage (1734 cm$^{-1}$, 1248 cm$^{-1}$), and total carbohydrate percentage (1205 cm$^{-1}$, 1111 cm$^{-1}$) than the CO. The abundance of guaiacyl-type lignin was detected in CO (862 cm$^{-1}$, 1516 cm$^{-1}$) with an overall increase in lignin content (1459 cm$^{-1}$), while CP showed more syringyl lignin and less total lignin compared with CO.

**Table 4.** FTIR peaks obtained and their assignments.

| Wave Number Range (cm$^{-1}$) | Samples and Their Obtained Peaks (cm$^{-1}$) | | | | | Generic Functional Group Assignment, Reference | Lignocellulose Specific Assignment |
|---|---|---|---|---|---|---|---|
| | CO | CP | AC | CL | LG | | |
| 3650–3600 | | | | | | Non-bonded free -OH stretching. [82] | |
| 3400–3200 | | | | | | Bonded -OH stretching. [82] | |
| | 3584 | 3559 | 3571 | 3586 | | Intramolecular hydrogen bond O(2)H-O(6). [83] | Cellulose |
| | 3475 | 3453 | 3448 | 3465 | | Intramolecular hydrogen bond O(2)H-O(6). [83] | Cellulose |
| | | | | | 3430 | -OH (bonded) stretching. [84] | Lignin * |
| | | 3358 | 3351 | 3355 | | Intramolecular hydrogen bond O(3)H-O(5), [83] | Cellulose |
| | 3179 | 3124 | 3112 | 3123 | | Intermolecular hydrogen bond O(6)H-O(3), [83] | Cellulose |
| 3000–2850 | | | | | | C-H stretching: Alkanes/O-H stretching carboxylic acid/Aldehyde. [85] | |
| 2970–2860 | | | | | | CH—stretching region (saturated aliphatic group frequencies). [86] | |
| | | | | | 2937 | C-H stretch methyl and methylene groups (2942 HW lignin, 2938 SW lignin). [87] | SW.Lignin |
| | 2886 | 2898 | 2904 | 2902 | | Symmetric C-H stretching. [84] | Cellulose * |
| | | | | | 2842 | C-H stretch O-CH$_3$ group. [87] | Lignin |
| 1780–1640 | | | | | | C=O stretching: Ester/Aldehyde/Ketone/Carboxylic acid; C=C stretching: Alkene [85] | |
| | 1731 | 1733 | | | | Ketone/Aldehyde C=O stretching (unconjugated) [88] | Hemicellulose * |
| | | | | 1711 | | Non–conjugated carbonyl [89] | Lignin |
| | 1643 | 1635 | 1639 | 1641 | 1643 | Intramolecular hydrogen bond/absorbed water/Aromatic ketones stretching [84] | |

**Table 4.** *Cont.*

| Wave Number Range (cm$^{-1}$) | Samples and Their Obtained Peaks (cm$^{-1}$) | | | | | Generic Functional Group Assignment, Reference | Lignocellulose Specific Assignment |
|---|---|---|---|---|---|---|---|
| | CO | CP | AC | CL | LG | | |
| 1600–1475 | | | | | | C=C stretching–skeletal vibration of phenolic compounds such as lignin, -CH$_2$ bend. [85] | |
| | 1606 | 1604 | | | | Aromatic skeleton vibration [87] | Lignin * (S > G; G-con. > G-eth.) |
| | | | | | 1598 | The aromatic ring (C=C), C=O stretching vibrations [64]. | Lignin * (S > G; G-con. > G-eth.) |
| | 1516 | 1516 | | | 1510 | Aromatic ring (C=C) stretching [64]. | Lignin * (G > S) |
| | 1456 | 1462 | | 1458 | 1464 | Asymmetric bending of CH$_3$ in methoxy groups//CH$_2$ bending vibration [88] | Lignin * (S > G), Cellulose, Hemicellulose |
| | 1425 | 1427 | 1429 | 1431 | | Scissoring motion of -CH$_2$ [60] | Cellulose-I * Crystallinity peak |
| | | | | | | O-CH$_3$ C-H deformation symmetric [87] | Lignin |
| | 1372 | 1374 | 1372 | 1372 | 1376 | Symmetric and asymmetric C-H deformation [85] | Cellulose, Hemicellulose, Lignin |
| | 1335 | | 1337 | 1337 | | C-H, -OH in-plane bending/weak C-O stretching [90] | Cellulose amorphous |
| | | | | | 1327 | Stretching of C-O in syringyl ring [91] | Lignin-S * |
| | | 1318 | 1316 | 1314 | | -CH$_2$ wagging [92] | Cellulose I crystalline |
| 1300–1000 | | | | | | C=O/C-O-C/C-O-H; Alcohols, ethers, esters, carboxylic acids, anhydrides [93] | |
| | | 1281 | 1281 | | | C-H bending [91] | Cellulose crystalline * |
| | | | | | 1269 | Aromatic ring vibration [85] | Lignin-G |
| | 1248 | 1251 | | | | C-O-C and C-O Stretching [94] | Hemicellulose * |
| | | | | | 1220 | C=O stretching of guaiacyl ring [95] | Lignin G |
| | 1205 | 1203 | 1201 | 1203 | | O-H in-plane bending [89] | Carbohydrates * |
| | 1158 | 1162 | 1164 | 1166 | | C-O-C stretching, Asymmetric stretching of C-O, C-C, O-H stretching of C-OH group [94] | Crystalline cellulose, β-glycosidic bond |
| | | | | | 1137 | C-H (aromatic) in-plane deformation, secondary alcohols, C-O stretch [59], | Lignin G |
| | 1111 | 1113 | 1113 | 1115 | | Asymmetric stretching of C-O-C; Cellulose characteristic peak [84] | Cellulose * |
| | | | | | 1082 | C-O deformation, secondary alcohol, an aliphatic ether [87] | Lignin |
| | 993 | 993 | 987 | 986 | | C-O and C-C, C-H bending or CH2 (amorphous band) stretching [96] | Cellulose |
| 1000–650 | | | | | | Out-of-plane bend Alkenes/Aromatics, aromatic C-H stretching [85] | |
| | 899 | 899 | 897 | 895 | | C-O-C stretching at β-1,4 glycosidic link [84] | Amorphous band * |
| | 862 | | | | 858 | C-H out of the plane in positions 2, 5, and 6 of G-ring [97] | Lignin-G |
| | | | | 814 | 817 | The vibration of mannan. CH out-of-plane bending in phenyl rings [98] | Glucomannan, Lignin G |
| | | | 714 | 714 | | Alcohol, OH out-of-plane bend. [99] | Cellulose Iβ * |
| | 668 | 668 | 668 | 668 | | -OH out-of-plane-bending [100] | Cellulose amorphous |
| | 607 | 617 | 619 | 617 | 617 | Alkyne C–H bend, Alcohol, OH out-of-plane bend [95] | Carbohydrates/Lignin |
| | 524 | 527 | 520 | 518 | 520 | C-O-C bending, C-C-C ring deform [101] | Cellulose, β-glycosidic bond |

SW: softwood; HW: hardwood; * characteristic peaks; G: guaiacyl; S: syringyl; G-con: condensed guaiacyl ring; G-eth: etherified guaiacyl ring.

In addition, the adsorbed water content was less in the case of equally dried CP compared with CO. These findings showed an overall increase in the carbohydrate to lignin ratio, hemicellulose to total carbohydrate ratio, and hemicellulose to lignin ratios in CP compared with that of CO (Table 5). The absence of lignin and hemicellulose peaks in the spectrum of AV and CL indicates their purity. The FTIR spectrum previously reported for the whole corncob was quite similar to that of both the CO and CP of this study [75]. The lignin to carbohydrate ratios previously reported were the same as that of CO, and these values were shown to get closer to that of CP when the corncob was pretreated with dilute acids and alkalis, proving the lignocellulosic construct of CP reported in this work [81]. The HBI value previously reported for the whole corncob is quite similar to that of the CO of this study and is reportedly decreased upon pretreatment [48]. The TCI, LOI, and CrI% values of a xylose-extracted corncob residue reported by Chi et al. [102] were slightly more than that of the CO in this work, indicating the decreased crystallinity of the biomass due to the presence of relatively amorphous constituents such as hemicellulose and lignin. On the other hand, the TCI and LOI values of the pure cellulose reference AC reported in the literature [103] are consistent with this work. All of the FTIR peaks of a whole corncob as reported by Zheng et al. [81] were also observed in the case of the CO. The S/G ratios of CO reported in this work are consistent with that of the whole corncob reported by HPLC [104] and NMR methods [105].

**Table 5.** Lignocellulose composition ratios measured by FTIR data.

| Ratio | Wave Number Range (cm$^{-1}$) | CO | CP | LG |
|---|---|---|---|---|
| S/G | 1462/1510–1508 | 1.34 | 1.38 | 0.52 |
| S/G | 1595/1510–1508 | 1.28 | 1.34 | 2.54 |
| LG/TC | 1510–1508/1374 | 1.03 | 0.71 | 8.75 |
| LG/TC | 1510–1508/1162 | 0.45 | 0.34 | N.A1 |
| LG/TC | 1510–1508/898 | 2.89 | 1.93 | N.A1 |
| XY/TC | 1734/1374 | 1.16 | 1.88 | N.A2 |
| XY/TC | 1734/1162 | 0.50 | 0.90 | N.A2 |
| LG/XY | 1510–1508/1734 | 0.88 | 0.37 | N.A1 |

S/G: syringyl/syringyl + guaiacyl ratio; LG/TC: lignin/total carbohydrate ratio; XY/TC: xylan/total carbohydrate ratio; LG/XY: lignin/xylan ratio; N.A1: lignin-related peaks are present but carbohydrate peaks are absent; N.A2: carbohydrate-related peaks are absent.

*3.7. XRD Analysis*

Diffractograms of the CO, CP, AC, and CL showed the lignocellulose characteristics of crystal lattice peaks with different intensities [106], such as (101) in the 2θ angle range of 14–15°, (10Ī) in the 16.5–17° range, (021) around 20.8°, (002) around 22.6°, and (040) around 34.3°. An amorphous characteristic plateau spanning between the peaks (10Ī) and (002) with its center around 18° was also observed. The results of crystallinity measurements by all four of the methods used were consistent (Table 6). The measured crystallinity of the samples was in the order of AC > CL > CO > CP. The results of Cra1% and Cra2% were similar for all samples. The method followed for the analysis of Cra2% was found to be advantageous to that of Cra1%, as the former can achieve the result without an additional step of normalization that could otherwise misinterpret the data (Figure 6, Table 6). The d-spacing of all samples was comparable (Table 6), whereas the crystallite sizes of the 002 lattice (L) of CO were the highest, and those of CP were the smallest. All results of AC and CL were similar. The observed differences between CO and CP strongly reflect the differences in their lignocellulosic construct (Table 6). The crystallinity (CrI%) and crystallite size (L) values reported for AC are consistent with the reported values in the literature [107]. Moreover, the difference between the values of CrI% and Crd% is consistent with the values reported in the literature for different types of cellulosic compounds [108]. The CrI values of the whole corncob previously reported were in the range of 35.19–39.2%; these values are almost half of that shown by CO in this work, proving the effect of separating amorphous CP from the whole corncob. Additionally, these works reported

the increase in the CrI of the corncob residue after removing its amorphous content (xylose or lignin) by the pretreatments employed [52]. Both the XRD (CrI%, Crd%, Cra1%, Cra2%) and FTIR (TCI, LOI, HBI) methods used for crystallinity measurement showed a lower crystallinity of CP compared with that of CO, AC, and CL, explaining the amorphous nature of CP due to its higher hemicellulose and syringyl lignin (Table 5). However, the CO showed slightly higher crystallinity than AC and CL in the FTIR measurement and a lower crystallinity in the XRD measurement. This observed difference in crystallinity among two different methods can be explained by two reasons: crystallinity measurement by FTIR methods is not absolute but is relative, and the readings are greatly influenced by the amorphous content (hemicellulose and lignin) of the sample [109]; and the XRD readings are dependent on crystallite size rather than particle size, thus the AC and CL having pure cellulose crystallite provided much sharper peaks than CO. The patterns of the FTIR, XRD, and TGA curves were consistent with that of the whole corncob reported [110]. The XRD plots of all the samples analyzed are given in the Supplementary Materials (Figure S13).

**Table 6.** Crystallinity measurements of samples by both the XRD and FTIR-based indices.

| Sample | XRD Analysis | | | | | | FTIR Analysis | | |
|---|---|---|---|---|---|---|---|---|---|
| | CrI% | Crd% | Cra1% | Cra2% | L | d | TCI | LOI | HBI |
| CO | 70.0 | 93.0 | 26.48 | 25.20 | 5.75 | 0.34 | 2.82 | 2.35 | 2.46 |
| CP | 31.0 | 73.0 | 20.06 | 23.84 | 2.94 | 0.41 | 1.47 | 0.87 | 2.03 |
| AC | 93.0 | 78.0 | 48.04 | 48.04 | 4.67 | 0.40 | 1.72 | 1.29 | 2.15 |
| CL | 91.0 | 77.0 | 44.28 | 36.01 | 4.73 | 0.39 | 1.8 | 0.96 | 1.89 |

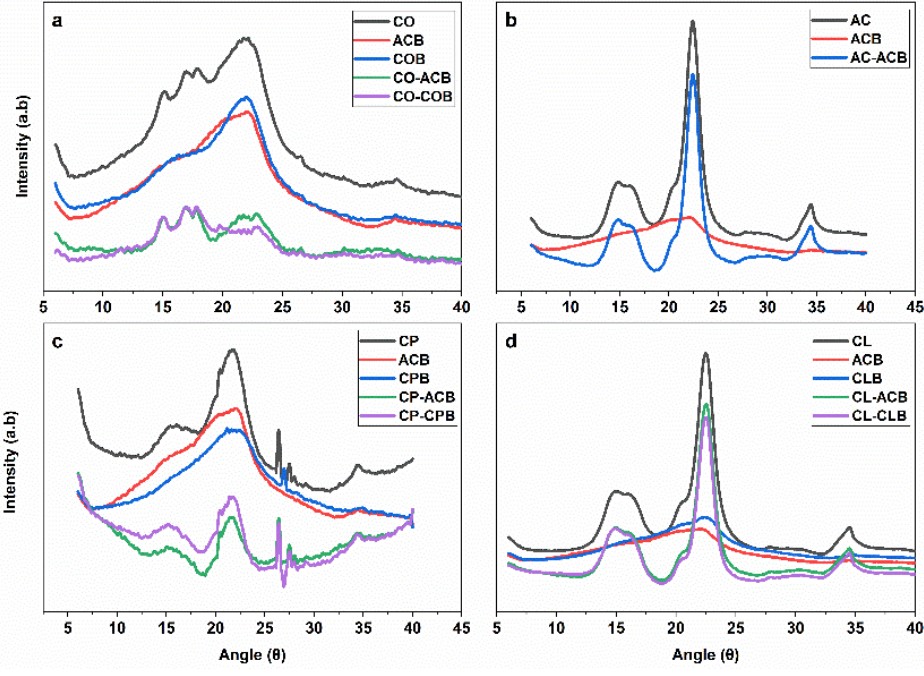

**Figure 6.** Amorphous contribution subtraction of XRD diffraction. Note: ACB, CLB, COB, and CPB are the diffraction patterns of the ball-milled AC, CL, CO, and CP, respectively; the negative sign indicates the diffraction of the sample after subtracting the diffraction of amorphous standards from it. For example, CO-ACB: diffraction of CO after subtracting amorphous contribution using diffraction of ACB; (**a–d**): Decrease in diffraction of around 18° and sharpening of the crystalline lattice by around 22° indicate the amorphous subtraction; (**a,c**): Diffraction patterns of CO and CP are significantly different, suggesting their varied crystallinities. Both COB- and CPB-subtracted samples showed slightly sharper patterns than that of ACB-subtracted samples; (**d**) CLB achieved a better amorphous subtraction than ACB.

*3.8. Enzymatic Saccharification of Untreated Corncob Samples*

A saccharification yield of 50–60% of the theoretical yield (TY) of CL and XY was obtained during the first 5 h of the incubation, which later gradually increased to 72.8% and 90.13%, respectively, after 40 h and 30 h. The saccharification of CP gradually increased and achieved a maximum yield close to that of controls, which was 70.57% of its TY at 50 h with cellulase and 88.70% of its TY at 50 h with xylanase. CO showed comparatively poor enzymatic saccharification susceptibility, showing no significant improvement from a minute saccharification yield of 15–18% of its TY obtained at the 10 h interval with both enzymes (Figure 7).

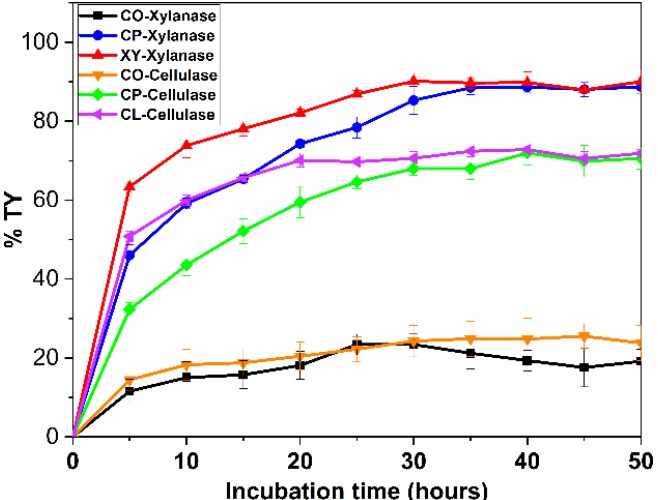

**Figure 7.** Enzymatic saccharification of the samples. Note: CO/CP/XY-xylanase: samples of CO, CP, or XY treated with xylanase; CO/CP/CL-cellulase: samples of CO, CP, or CL treated with cellulase; %TY: percentage of the theoretical yield (saccharification) achieved.

The maximum TY of CO with xylanase was around 26% of the reference XY, where CP achieved 98.4% of it. The maximum TY of CO with cellulase was around 35% of reference CL, where CP has achieved 98.8% of it. These results are perfectly correlated with the chemical and physical characterization of the respective corncob anatomical portions. As per the NREL method of composition analysis, CP on average showed a 20.7% lower lignin percentage along with a higher percentage of cellulose, hemicellulose, and extractives (6.8%, 1.9%, and 21.4%, respectively). A similar difference was observed from other composition analysis methods reported in this work. In addition, the S/G and XY/TC ratios of CP were 3.8%, which was 67.4% higher; the LG/TC and LG/XY ratios of CP were 31.8% and 57.9% lower than that of CO, respectively. The crystallinity values of the CP measured by both the XRD (CrI%, Crd%, Cra1%, and Cra2%) and FTIR (TCI, LOI, and HBI) methods were 55.7%, 21.5%, 24.2%, 5.3%, 47.8%, 62.9%, and 17.4% lower than that of CO, respectively. A huge contrast observed in enzymatic saccharification susceptibility of untreated CO and CP can be essentially attributed to their chemical compositional differences, especially to their lignin to carbohydrate ratios and to their differences in crystallinity. Although CP has a slightly higher syringyl percentage than CO, the S/G ratio appears to be a comparatively minor deciding factor for their saccharification susceptibilities.

The saccharification profile of CO in this study is similar to that of the whole corncob without pretreatment as previously reported by many other researchers as a control in their respective studies [5,111]. Whole corncob ground to a similar mesh as that of the CO in this study reportedly achieved a similar saccharification yield by the first 10 h interval and was unchanged thereafter using cellulase of the same make as that used in this study [112] and when using cellulase procured from a different manufacturer [113]. Similar yields and patterns were reported even when the cellulase activity was complimented with β-glucosidase [5,114]. On the other hand, many works reported enzymatic production

of xylooligosaccharides from pretreated whole corncob, either by in-house-produced xylanases [115], or with commercial xylanases [116]; however, none of these studies showed the effect of xylanases on an untreated corncob. Nevertheless, we found a report where the whole corncob without any chemical pretreatment was used as a control for an in-house-produced *T. viride*-derived xylanase; the enzyme activity profile reported for the untreated whole corncob was similar to that of the CO in this study, but the peak activity was achieved at 48 h of incubation. [117]. However, we did not find any work reporting the saccharification of individual anatomical portions of corncob to date.

## 4. Conclusions

The comprehensive characterization of the corncob anatomical portions revealed the striking morphological, structural, and chemical differences among the outer (CO) and pith (CP) sections of each corn variety studied; at the same time, there are no significant differences among the same anatomical portion in different corn varieties. Most of the characteristics of the CO were similar to that of whole corncob characteristics vividly reported in the literature, whereas CP showed unique characteristics, such as lower lignin, protein, and ash contents with an improved xylan and cellulose content. NIR-PLS calibration models along with Savitzky-Golay smoothing of the spectra are proven to be the fittest for the rapid composition analysis of all the biomass components. Both the FTIR and XRD analyses showed that CO is more crystalline than CP, and the thermal stability of CP was found to be lower than that of CO. All of these compositional and physical differences led to enhanced enzymatic saccharification of CP by both cellulase and xylanases, which was equal to that of the pure cellulose (AC), and xylan (XY) references. Thus, we propose a tailored enzymatic production of xylooligosaccharides from CPs without pretreatment along with a separate valorization of CO to achieve an economical biorefinery output from the corncob feedstock. However, the techno-economic evaluation of the proposed process must be carried out to assess the viability of the process given the newly included step of biomass anatomical segregation.

**Supplementary Materials:** The following supporting information can be downloaded at: https://www.mdpi.com/article/10.3390/fermentation8120704/s1, Figure S1: Carbohydrate calibration-1; Figure S2: Carbohydrate calibration-2; Figure S3: Carbohydrate calibration-3; Figure S4: Acetate calibration; Figure S5: Structural carbohydrates-CO; Figure S6: Structural carbohydrates-CP; Figure S7: Structural carbohydrates-AC; Figure S8: Structural carbohydrates-CL; Figure S9: Acetate-CO; Figure S10: Acetate-CP; Figure S11: Acetate-AC; Figure S12: Acetate-CL; Figure S13: XRD profiles of the samples.

**Author Contributions:** P.K.G.: Conceptualization, methodology, investigation, formal analysis, data curation, writing—original draft preparation; M.L.C.: writing—original draft preparation, investigation, formal analysis, reviewing and editing; A.P.G.: investigation, validation, visualization; N.P.P.P.: visualization, reviewing and editing; S.K.: formal analysis, reviewing and editing; A.B.: investigation, formal analysis; S.R.A.: investigation; R.R.B.: conceptualization, editing, supervision, resources, funding acquisition, project administration. R.K.B.: reviewing and editing. All authors have read and agreed to the published version of the manuscript.

**Funding:** This research work was supported by the Department of Science & Technology-Science and Engineering Research Board (DST-SERB), India, for the Early Career Research Grant Reference No. ECR/2015/000076.

**Data Availability Statement:** The data published in this article is not archived in any public databases.

**Acknowledgments:** The corresponding author R.R.B. acknowledges N.V. Ramana Rao, N.I.T., Warangal, Telangana, India, for his encouragement and constant support to carry out this work. Authors P.K.G. and M.L.C. acknowledge N.I.T., Warangal, Telangana, India for the institute fellowship and constant support provided to carry out this work; the authors also acknowledge S. Sakthivel, Scientist-F and head of the Center of Solar Energy Materials (CSEM), ARCI, Balapur, Hyderabad, for offering assistance during the characterization.

**Conflicts of Interest:** The authors declare that they have no known competing financial or non-financial interests nor any personal relationships that could have directly or indirectly appeared to influence the work reported in this paper.

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
