# Peer review of "A New Insight into the Composition and Physical Characteristics of Corncob—Substantiating Its Potential for Tailored Biorefinery Objectives"

_fermentation, doi:10.3390/fermentation8120704_

Round 1

Reviewer 1 Report

The paper provides a diligent and novel study of the composition and properties of two different anatomical parts of corn cobs with high relevance for biorefinery, esp. enzymatic hydrolysis & fermentation.

Some minor revision is required:

1) lines 69 and 122: "cellulose" not "cellulase"? Please check

2) lines 136-137: meaning of the following phrase is not clear: ".....and showed the decrease in biomass recalcitrance up on pretreatment promotes the enzymatic saccharification...."

3) lines 333-334: could you please provide a short description of the "micro-DNS-Assay"

4) Figure 2: please add the magnifications in the figure title; the fonts of the bars in the images are too small

5) Table 1: please provide more space between the columns

6) R&D-part: I´d suggest to present first the Van Soest, then the NIRS - data - for easier comparison with the results by NREL - methods, i.e. change 3.3. and 3.4

7) Table 1 and Table 2: there are considerable differences in lignin findings in nearly all samples, as well as in xylan/hemicellulose findings in CL and AC, between van Soest and NREL methods. Esp. the xylan and mannan detected by NREL methods in AC and CL do not really fit to Van Soest and TGA analysis. Could you please give a short discussion on this.

8) lines 429, 417, 442, 444, 484, 520: I suppose the figure references are not correct. Please check.

9) line 523:I suppose you mean "CL", not "AC" - see fig. 6

10) part 3.3. Could you please give some more detailed information on the NIRS results, e.g. which data did you use for the calibration models?

Author Response

Reviewer 1: The paper provides a diligent and novel study of the composition and properties of two different anatomical parts of corn cobs with high relevance for biorefinery, esp. enzymatic hydrolysis & fermentation. Some minor revision is required:

Comments

Response

1)      Lines 69 and 122: "cellulose" not "cellulase"? Please check

In Line 69 it was a typographical error and is corrected as cellulose, but in line 122 it is cellulase, we slightly changed the sentence construct there to make it more meaningful.

2) Lines 136-137: meaning of the following phrase is not clear: ".....and showed the decrease in biomass recalcitrance up on pretreatment promotes the enzymatic saccharification...."

The sentence has been reframed to make it more significant.

3) Lines 333-334: could you please provide a short description of the "micro-DNS-Assay"

The micro-DNS assay performed in this study was based on a well-known macro-DNS assay proposed by T. K. Ghose [1], a short description of this assay has been added in the manuscript.

4) Figure 2: please add the magnifications in the figure title; the fonts of the bars in the images are too small

The fonts presented earlier were instrument generated hence can’t be magnified as they are. Now in the revised figure-2, we edited the image fonts to magnify the important details.

5) Table 1: please provide more space between the columns

Table -1 is too wide to be published horizontally in a portrait mode page, either the table needs to be turned vertically or the page has to be horizontal. In this revised manuscript we turned the page to horizontal mode, and the publishing house can take the final decision on the best way to fit it.

6) R&D-part: I´d suggest presenting first the Van Soest, then the NIRS - data - for easier comparison with the results by NREL - methods, i.e. change 3.3. and 3.4

Said sections were interchanged as suggested, both in the methods and results sections.

7) Table 1 and Table 2: there are considerable differences in lignin findings in nearly all samples, as well as in xylan/hemicellulose findings in CL and AC, between van Soest and NREL methods. Esp. the xylan and mannan detected by NREL methods in AC and CL do not really fit to Van Soest and TGA analysis. Could you please give a short discussion on this.

It is a common observation that has been reported previously with other biomass types as well. The compositional analysis methods used in this study do not give the same exact result for any biomass, owing to the differences in their methodologies. These methods are relative but none of them are absolute, yet they are alternatively used in biorefinery studies. For example, NREL method involves complete acid hydrolysis of extractive free samples followed by HPLC analysis of individual monomers (hemicellulose is expressed as individual components xylan, arabinan, and mannan), whereas, Vansoest method involves detergent solubilization of structural and non-structural components of biomass followed by gravimetric analysis. ADF solution might have solubilized some amount of cellulose too, hence the hemicellulose concentrations were higher than cellulose. Nevertheless, we can find a higher hemicellulose content in CP than CO. Although NREL method has gained popularity these days, still other methods are also being used and cited. Hence in this work, we tried to compile the compositions obtained from all the methods in one place as we are reporting it for the first time for different anatomical portions of the corncob. 

The Physico-chemical construct of corncob outer (CO) is quite similar to that of the whole corncob, hence we compared our respective results with that of the whole corncob reported in the literature. In addition, the reason for variation among crystallinity measurements by FTIR and XRD techniques  is discussed in detail in section 3.7

8) Lines 417, 429, 442, 444, 484, 520: I suppose the figure references are not correct. Please check.

It was a mistake that occurred due to the error in the auto-captioning of figures. Now it is rectified.

9) Line 523:I suppose you mean "CL", not "AC" - see fig. 6

Yes, it was an error. Changed “AC” to “CL”

10) Part 3.3. Could you please give some more detailed information on the NIRS results, e.g. which data did you use for the calibration models?

The details of NIR data acquisition and its segregation to make calibration and validation sets for PLS analysis were given in section 2.5. We now included NIR spectrum figure in supplementary data for your perusal.  Please specify if any other information is needed.

Reviewer 2 Report

The paper describes fractionation of an abundant agro-waste (corncob) for particular application in biorefinery purposes. The article is interesting, and analysis is exhaustive. Minor changes can improve the article.

L22: Remove ‘each’. Also mention if these were physically segregated or any chemical method was applied.

L55: Correct in text citations. Check it throughout the paper

At present introduction looks messy. It is suggested to write a separate paragraph for pretreatment and its cost etc. Make it a brief section as the work is not pertaining to pretreatment. The work related to fractionation of corncob is relevant and that can be written separately. Likewise, the effect of constituent of lignin on biorefinery processes should be made to a separate paragraph. Moreover, the effect of particle size and other variables on enzymatic saccharification should also be referred separately.

L160: Write units correctly

L220: Check if its NDF or PDF

L320: Italicize the names of the organism

L401: Look for the unit.

The discussion needs improvement. It would be better to emphasize over the significance of the results, particularly in relation to practical implications.

Multiple references for a single statement do not need to be so extensive. The list of references can easily be trimmed to one third of the current citations.

Author Response

Reviewer 2: The paper describes fractionation of an abundant agro-waste (corncob) for particular application in biorefinery purposes. The article is interesting, and analysis is exhaustive. Minor changes can improve the article.

Comments

Response

L22: Remove ‘each’. Also mention if these were physically segregated or any chemical method was applied.

Removed the word “each” and included the term “physically” to indicate the process of segregation precisely.

L55: Correct in text citations. Check it throughout the paper

Multiple citations are combined.

At present introduction looks messy. It is suggested to write a separate paragraph for pretreatment and its cost etc. Make it a brief section as the work is not pertaining to pretreatment. The work related to fractionation of corncob is relevant and that can be written separately. Likewise, the effect of constituent of lignin on biorefinery processes should be made to a separate paragraph. Moreover, the effect of particle size and other variables on enzymatic saccharification should also be referred separately.

A paragraph on biomass pretreatment and its effect on overall process economics is newly included.

There is no previously reported literature available for corncob fractionation. We are the first to report it, this is the novelty of this work.                                                             

The effect of the lignin content of biomass on its recalcitrance was already briefly discussed, please suggest if you need any additional information

The effect of particle size on the conversion is newly included, and the effect of remaining physical parameters was already discussed.

L160: Write units correctly

All units has be corrected 

L220: Check if its NDF or PDF

Yes it was an error and has been corrected 

L320: Italicize the names of the organism

Amended

L401: Look for the unit.

The said unit is corrected throughout the manuscript.

The discussion needs improvement. It would be better to emphasize over the significance of the results, particularly in relation to practical implications.

We included an elaborate discussion of results concerning the relationship between the physical and chemical construct of the biomass in the study. And also tried to compare our results with the reported literature as much as possible. As per the practical implication, the very objective of this paper is to assess the readiness of corncob pith for enzymatic saccharification without a pretreatment and proved that it can be saccharified without a pretreatment. Yet these findings need to be further techno-economically evaluated for the viability of the process given an additional step of biomass-anatomical segregation included. We included a few lines in the conclusion to make it more practical.

Multiple references for a single statement do not need to be so extensive. The list of references can easily be trimmed to one third of the current citations.

Multiple references were given wherever the variation in reported literature or the contradictory findings were observed. For example, significant variations in lignocellulose compositions for the whole corncob were reported, and also varied opinions on the effect of the physicochemical properties of biomass over its bioconversion efficiency were reported. However, the overall citation has been reviewed carefully and only significant references have been cited in the manuscript.

Round 2

Reviewer 2 Report

All the comments have been addressed. Just a reminder to split introduction in to paragraphs